# NLRX1 Mediates the Disruption of Intestinal Mucosal Function Caused by Porcine Astrovirus Infection via the Extracellular Regulated Protein Kinases/Myosin Light–Chain Kinase (ERK/MLCK) Pathway

**DOI:** 10.3390/cells13110913

**Published:** 2024-05-25

**Authors:** Jie Tao, Jinghua Cheng, Ying Shi, Benqiang Li, Pan Tang, Jiajie Jiao, Huili Liu

**Affiliations:** 1Institute of Animal Husbandry and Veterinary Medicine, Shanghai Academy of Agricultural Sciences, Shanghai 201106, China; taojie@saas.sh.cn (J.T.); zero5cheng@saas.sh.cn (J.C.); shiying@saas.sh.cn (Y.S.); huteng2010@saas.sh.cn (B.L.); tpvip2023@saas.sh.cn (P.T.); jiaojiajie@saas.sh.cn (J.J.); 2Shanghai Key Laboratory of Agricultural Genetic Breeding, Shanghai 201106, China; 3Shanghai Engineering Research Center of Pig Breeding, Shanghai 201302, China

**Keywords:** *porcine astrovirus*, NLRX1, mitophagy, ERK/MLCK, intestinal mucosal barrier

## Abstract

*Porcine astrovirus* (*PAstV*) has a potential zoonotic risk, with a high proportion of co-infection occurring with *porcine epidemic diarrhea virus* (*PEDV*) and other diarrheal pathogens. Despite its high prevalence, the cellular mechanism of *PAstV* pathogenesis is ill–defined. Previous proteomics analyses have revealed that the differentially expressed protein NOD–like receptor X1 (NLRX1) located in the mitochondria participates in several important antiviral signaling pathways in *PAstV–4* infection, which are closely related to mitophagy. In this study, we confirmed that *PAstV–4* infection significantly up-regulated NLRX1 and mitophagy in Caco–2 cells, while the silencing of NLRX1 or the treatment of mitophagy inhibitor 3–MA inhibited PAstV–4 replication. Additionally, *PAstV–4* infection triggered the activation of the extracellular regulated protein kinases/ myosin light-chain kinase (ERK/MLCK) pathway, followed by the down-regulation of tight–junction proteins (occludin and ZO–1) as well as MUC–2 expression. The silencing of NLRX1 or the treatment of 3–MA inhibited myosin light-chain (MLC) phosphorylation and up-regulated occludin and ZO–1 proteins. Treatment of the ERK inhibitor PD98059 also inhibited MLC phosphorylation, while MLCK inhibitor ML-7 mitigated the down-regulation of mucosa-related protein expression induced by *PAstV–4* infection. Yet, adding PD98059 or ML–7 did not affect NLRX1 expression. In summary, this study preliminarily explains that NLRX1 plays an important role in the disruption of intestinal mucosal function triggered by *PAstV–4* infection via the ERK/MLC pathway. It will be helpful for further antiviral drug target screening and disease therapy.

## 1. Introduction

*Astrovirus* (*AstV*) belongs to the family *Astroviridae*, including *Mamastrovirus* (33 species) and *Avastrovirus* (7 species) [1]. *AstV* is a potentially zoonotic pathogen that is widely distributed around the world and has been found to infect humans, dogs, cats, pigs, sheep, cattle, mink, deer, bats, dolphins, turkeys, ducks, 31 species of mammals, and 6 species of poultry [2]. This diarrheal disease ranks as the second leading cause of death among children under 5 years of age, accounting for approximately 2 billion cases worldwide annually. In China, the rate of diarrheal disease in children is reported to be 1.9 times per person per year [3]. *Human astrovirus* (*HAstV*) is recognized as the second leading cause of gastroenteric diarrhea in humans [4]. *Porcine astrovirus* (*PAstV*), which belongs to the *Mamastrovirus* genus, was first detected by electron microscopy in piglet diarrhea samples in 1980 [5]. However, the first molecular characterization of *PAstV* was not reported until 2001 [6]. To date, five genotypes of *PAstV* (*PAstV–1* to *PAstV–5*) with different prevalences have been identified worldwide. In China, the positive rates of *PAstV* in East China, South China, North China, Central China, Northeast China, and Southwest China are 28.91%, 31.3%, 6.03%, 41.04%, 30.5%, and 32.3%, respectively [7]. *PAstV* exhibits diverse clinical symptoms due to the presence of different genotypes characterised by significant genetic variations [8]. *PAstV–3* can damage the central nervous system (CNS), while the other four genotypes are associated with diarrhea [9,10]. Statistics show that *PAstV–2* (81.48%) is the most predominant genotype in China, followed by *PAstV–4* (11.11%) and *PAstV–5* (7.41%) [11,12]. However, the prevalence ratio of *PAstV–4* is highest in European countries (70.4%) and other Asian countries, including Korea (88.46%), Thailand (92.0%), and India (95.65%) [13,14,15]. Additionally, *PAstVs* frequently coexist with other diarrhea-inducing diseases, such as *coronaviruses*, *caliciviruses*, and *sapelovirus* [12,16,17,18]. It is speculated that *PAstV* may be an important pathogen involved in the overwhelming majority of co–infection cases. Despite its prevalence, little is known about the mechanism by which *astrovirus* causes diarrhea. Hence, it is crucial to focus on research regarding the pathogenesis of *PAstV*, aiming to identify potential antiviral drug targets. This effort could offer new strategies for treating both human and animal gastroenteritis.

The intestinal mucosal barrier serves as the primary defense mechanism against foreign pathogens in the body. When this barrier is compromised, it leads to increased intestinal permeability, disrupting the intestinal barrier [19]. Ultimately, the compromise of the intestinal mucosal barrier initiates an inflammatory response, contributing to the onset of intestinal diseases [20]. It has been reported that granulosomal dysfunction can impair intestinal epithelial cell function integrity, culminating in gastroenteritis disease [21]. It has been reported that *AstV* only replicates in the intestine, possibly because it needs to change the permeability of the intestinal epithelial cell barrier through capsid proteins, disrupt intestinal tight connections, and expose surface receptors to achieve invasion. The *HAstV–1* infection of Caco–2 cells increases epithelial cell permeability by decreasing the occlusion protein and junction complex, and cadherin recombination. But the specific mechanism of *PAstV* is not clear, and different species of *AstVs* have great differences in pathogenicity [22].

Our previous research revealed that PAstV–4 infection in PK15 cells induces mitochondrial damage. Among the differential proteins, NLRX1, a member of the Nod-like receptor (NLR) family, is located in the mitochondria and is highly expressed in the intestine, and is known to modulate ROS production, mitochondrial damage, autophagy, and apoptosis [23]. This might suggest a potentially significant role for NLRX1 in the context of *PAstV–4* infection. Hence, in this study, we examined the impact of NLRX1 on *PAstV–4* replication and further investigated its regulatory role in disrupting intestinal mucosal barrier function induced by *PAstV–4* infection. Our findings indicated that NLRX1 could enhance the replication of *PAstV–4*, and the silencing of NLRX1 alleviated the down-regulation of tight-junction protein in intestinal epithelial cells through the ERK/MLCK pathway. In conclusion, the data in this study revealed that NLRX1 plays a vital role in destruction of the intestinal mucosal barrier induced by *PAstV–4* infection via mitophagy. Further, we explores whether this phenomenon is consistent in four other genotypes.

## 2. Materials and Methods

### 2.1. Cells, Viruses, and Antibodies

The human colon adenocarcinoma cell line Caco–2, provided by Prof. Guoqiang Zhu from Yang Zhou University, was cultured in Dulbecco’s modified Eagle’s medium (Gibco, Carlsbad, CA, USA) supplemented with 10% fetal bovine serum (Gibco, Carlsbad, CA, USA) within a humidified atmosphere containing 5% CO_2_ at 37 °C. The *porcine astrovirus* type 4 stain *PAstV/SH/2022/CM1*, identified in our laboratory, was propagated in Caco-2 cells with 15 μg/mL Pancreatin. The viral titer was determined by measuring the 50% tissue culture infective dose (TCID_50_). The mouse monoclonal antibody against *PAstV* was obtained from LanDu Biotechnology (Binzhou, China). Additionally, the anti–NLRX1 polyclonal antibody was received from Proteintech (Wuhan, China), and the anti–HA polyclonal antibody, anti–*β–action* antibody (Abcam, Cambridge, UK), HRP-conjugated anti-mouse IgG, and FITC–conjugated anti–mouse IgG (Abcam, Cambridge, UK) were utilized following the instructions of the manufacturers. 

### 2.2. Transfection and Silencing of the NLRX1 Gene Using siRNA

For siRNA knockdown experiments, Caco–2 cells were seeded in 6-well plates and transfected twice with 25 nM of the designated siRNAs over 48 h using Lipofectamine 3000 (Invitrogen, Carlsbad, CA, USA). The siRNA employed were as follows: siRNA/NLRX1–1, 5′–UUGUCAAUCUGCUGCGCAA–3′; siRNA/NLRX1–2, 5′–GUGCUGGGCUUGCGGAAGA–3′; siRNA/NLRX1–3, 5′–GCAUGUCUUCCGCCGGGAU–3′; negative control siRNA, 5′–UUCUCCGAACGUGUCACGUTT–3′. 

### 2.3. TCID_50_ for PAstV

*PAstV* titers were assessed through an indirect fluorescence assay using the Reed–Muench method. Caco–2 cells were seeded in 96–well plates, and eight replicates of serial 10–fold dilutions of *PAstV* were inoculated. After 96 h, an indirect fluorescence assay was conducted to calculate the TCID_50_ using the *PAstV* specific antibody [24]. 

### 2.4. Quantitative Real-Time PCR (qPCR)

At the specified time points, total RNA was extracted using TRIzol reagent (Invitrogen, Carlsbad, CA, USA). Subsequently, reverse transcription was carried out utilizing a PrimeScript^®^ RT reagent Kit with a gDNA Eraser (Takara Bio, Otsu, Japan). Following reverse transcription, quantitative PCR (qPCR) was conducted using SYBR Green Real–Time PCR Master Mix (Takara Bio, Otsu, Japan) and an ABI7500 system (ABI, Madison, WI, USA). The amplification conditions included initial denaturation at 95 °C for 30 s, followed by 40 cycles of denaturation at 95 °C for 5 s, annealing at 60 °C for 30 s, and extension at 95 °C for 15 s. The *β–actin* gene served as an internal standard, and the results were calculated utilizing the 2^−ΔΔCt^ method [25]. The primer sequences employed for amplification were as follows: *occludin* (F: 5′–TCCAACGGGAAAGTGAACGA–3′; R: 5–GTGGATATTCCCTGATCCAGTCTT–3′), *ZO-1* (F: 5′–AAGGTAAAGTCTGCTGAGGCTGAA–3′; R: 5′–GACACTGAATTACCTTCGCCG-3′), *MUC–2* (F: 5′–GTCGAGTACATCCTGCTGACG–3′; R: 5′–GAGTCCTCTCTGTTTCCACACG–3′), *NLRX1*(F: 5′–CAGACCCTCACAAGCATCTA–3’; R: 5′–CACGGACATCCTCTTCAGA–3′), *β-actin* (F: 5′–TGGGTCAGAAGGACTCCTATG–3′; R: 5′–CAGGCAGCTCATAGCTCTTCT–3′). 

### 2.5. Western Blotting Analysis

Cells cultured in 60 mm dishes were prepared by adding 200 μL of 2 × lysis buffer A (65 mM Tris–HCL, 4% SDS, 3% DL–dithiothreitol, and 40% glycerol) and incubated for 30 min on ice. Subsequently, the samples were mixed with 5×SDS buffer and boiled for 10 min. The proteins were separated by 12% SDS–PAGE and electroblotted onto a polyvinylidene difluoride (PVDF) membrane (Millipore, Billerica, MA, USA) using eBlot L1 (GenScriptTM, Nanjing, China). After blocking for 1 h at room temperature with Tris–buffered saline containing 5% skim milk, the membranes were incubated overnight at 4 °C with the specified primary antibodies. This was followed by incubation with HRP–conjugated secondary antibodies at room temperature for 1 h. Finally, the proteins were visualized via staining using enhanced chemiluminescence detection kits (Thermo Fisher Scientific, Waltham, MA, USA). 

### 2.6. Statistical Analysis

The values are presented as mean ± standard deviation (SD). Data were analyzed using Student’s *t*-test and processed using SPSS. *p* values less than 0.05 were considered statistically significant.

## 3. Results

### 3.1. Down-Regulation of Tight-Junction Proteins in Caco–2 Cells Following PAstV–4 Infection

To investigate the impact of *PAstV* infection on the mucosal barrier function of intestinal epithelial cells, the susceptibility of *PAstV–4* to human colon cancer epithelial cells (Caco–2) was initially assessed. The results confirmed the successful proliferation of the *PAstV/SH/2022/CM1* strain on Caco–2 cells, with a viable titer of 10^5.23^TCID_50_/0.1 mL (Figure 1A). Subsequently, the effect of *PAstV* infection on epithelial mucosal barrier function was assessed. The results demonstrated down-regulation in the transcription and expression of the MUC–2 protein and the tight-junction proteins of occludin and ZO–1 upon PAstV–4 infection (Figure 1B). Furthermore, the degree of inhibition increased with an increasing infection dose of *PAstV–4* (Figure 1C).

### 3.2. Up-Regulation of NLRX1 Expression Induced by PAstV Infection

Building upon our prior findings indicating that *PAstV/SH/2022/CM1* infection up-regulated the expression of NLRX1 in PK15 cells, we sought to confirm this result in the intestinal epithelial cells during *PAstV* infection. To achieve this, caco−2 cells were infected with *PAstV−4* at an MOI of 1. Real-time PCR analysis confirmed the significant up-regulation of NLRX1 expression at 24 h post-infection (Figure 2A). Notably, UV−inactivated *PAstV*−*4* failed to induce NLRX1 mRNA expression, suggesting that the up-regulation of NLRX1 depends on viral replication (Figure 2A). Furthermore, the results revealed that *PAstV*−*4* up–regulated NLRX1 dose-dependently (Figure 2B).

### 3.3. PAstV–4 Replication Augmented by NLRX1 Knockdown in Caco−2 Cells

To investigate the impact of the endogenous NLRX1 on *PAstV−4* replication, three pairs of siRNA duplexes targeting NRLX1 were separately transfected into Caco-2 cells. The knockdown efficiency of these siRNAs was assessed through real-time PCR, revealing that siRNA/NLRX1−2 achieved the highest efficiency (Figure 2C). Western blot analysis confirmed that siRNA/NLRX1−2 significantly reduced the level of NLRX1 protein by approximately 89% compared to control cells transfected with NC siRNA (Figure 2D). Subsequently, siRNA/NLRX1−2 was transfected into Caco−2 cells, followed by *PAstV–4* infection, which decreased *PAstV–4* replication within Caco−2 cells (Figure 2E). This suggests that the endogenous NLRX1 can potentially promote the replication of *PAstV–4* in Caco−2 cells. *PAstV−4* infection also resulted in the up-regulation of LC3II proteins, and this effect was inhibited by treatment with siRNA/NLRX1 (Figure 3A). Additionally, treatment with 20 μM 3−MA (mitophagy inhibitor) decreased the viral titer of *PAstV* (Figure 3B) while not affecting NRLX1 expression (Figure 3C,D). We speculated that *PAstV–4* infection activated mitochondrial autophagy to resist host innate immunity through the up-regulation of NLRX1, thereby promoting its replication.

### 3.4. Disruption of Caco−2 Mucosal Barrier through ERK/MLC Pathway Activation by NLRX1

The ERK/MLC pathway is crucially associated with intestinal mucosal barrier function. Following infection with *PAstV−4* (1.0 MOI) for 24 h, increased p38, p−ERK, and p−MLC levels were observed through Western blot analysis. Moreover, the knockdown of NLRX1 hindered the expression of p−ERK and p−MLC induced by *PAstV−4* infection (Figure 4A). Moreover, the treatment with 20 μM PD98059 (ERK inhibitor) decreased the expression of p−MLC. In contrast, NLRX1 expression remained unaffected (Figure 4B). Subsequently, the impact of mitophagy on the ERK/MLCK pathway was assessed. The results indicated that treatment with 3−MA down-regulates ERK and MLC phosphorylation and up-regulated the expression of occludin and ZO−1. This suggests that NLRX1 is implicated in regulating the activation of the ERK/MLCK pathway through mitophagy. 

To investigate the role of NLRX1 in regulating intestinal mucosal barrier function, siRNA/NLRX1−2 was transfected into Caco−2 cells, followed by *PAstV−4* infection. Subsequently, the expression levels of MUC−2, occludin, and ZO−1 were assessed using Western blot analysis. The results demonstrated that the down-regulation of the three proteins induced by *PAstV−4* infection was alleviated (Figure 5A). Furthermore, treatment with 25 μM ML−7 (MLCK inhibitor) exhibited this phenomenon while not affecting NLRX1 expression (Figure 5B,C). MLCK is responsible for the phosphorylation of MLC. This suggests that NRLX1 might regulate mucosal barrier function through the ERK/MLCK pathway during *PAstV−4* infection.

## 4. Discussion

*AstV* is recognized as a potential zoonotic pathogen, with reported infections in thirty−one species of mammals and six poultry species, leading to symptoms such as diarrhea and neurological issues [26]. In particular, *PAstV* primarily induces symptoms like diarrhea, vomiting, and anorexia in piglets. Diarrheal pig feces are characterised by a mushy consistency or yellow water, with pigs exhibiting clinical symptoms similar to those caused by the *porcine epidemic diarrhea virus* (*PEDV*) [12,27]. Furthermore, synergistic effects with *PEDV* may exist, emphasizing the need for careful attention to such co−infections. It is necessary to first study the mechanism of *PAstV* infection before studying its synergistic pathogenesis. Our previous study discovered that the NLRX1 protein, predominantly in mitochondria, exhibited significant up−regulation following *PAstV*−*4* infection. This up−regulation was associated with the mitochondrial autophagy pathway, indicating a potentially crucial role for NLRX1 in the context of *PAstV* infection. Hence, this study aimed to improve our understanding of the impact and regulation of the NLRX1 protein on *PAstV* infection.

NLRX1 is a modulator of pathogen-associated molecular pattern receptors and identifies a key therapeutic target for enhancing antiviral responses [28]. Li et al. found that NLRX1 regulated mitophagy via the FUNDC1−NIPSNAP1/NIPSNAP2 signaling pathway in intestinal ischemic reperfusion (IR) injury [29]. Based on the property of regulating mitophagy, NLRX1 was found to facilitate the replication of *human immunodeficiency virus 1* (*HIV−1*), *herpes simplex virus 1* (*HSV−1*), and *hepatitis C virus* (*HCV*), whereas it restricts *influenza A virus* (*IAV*), *hepatitis A virus* (*HAV*), and *porcine reproductive and respiratory syndrome virus* (*PRRSV*) replication [30]. Until now, the specific role of NLRX1 in *AstVs* infection was scarcely known. In this study, we found that *PAstV−4* infection up−regulated the expression of NLRX1 and LC3II, while silencing the expression of NLRX1 inhibited *PAstV* replication and mitophagy. It was speculated that *PAstV−4* activated mitophagy to resist host innate immunity through the up-regulation of NLRX1, thereby promoting viral replication. Several studies have confirmed the close relationship between intestinal barrier injury and mitophagy [31,32]. However, there are no report about NLRX1 being involved in this. This study first explored the role of NLRX1 on intestinal mucosal barrier injury induced by *PAstV−4* infection.

In *astrovirus* infection, destruction of the intestinal epithelium and the stimulation of an inflammatory response are not the main mechanisms that cause diarrhea [33]. Astrovirus increased barrier permeability in a Caco−2 cells, correlated with disruption of the tight-junction protein occludin, and decreased the number of actin stress fibers [34]. Indeed, the impairment of tight junctions and adhesive junction structures has been established as a contributing factor in the pathogenesis of *PEDV* and various other pathogens [35,36,37]. Oral administration of the *turkey astrovirus 2* (*TAstV−2*) structural (capsid) protein induced acute diarrhea, increasing barrier permeability [38]. *HAstV* infection increased the permeability of the endothelial cell barrier through the activation of MLC phosphorylation [39]. However, the effect of *PAstV* on the intestinal mucosal barrier remains unknown. Fang et al. reported that seven−day−old suckling piglets being infected with *PAstV-GX1* (*PAstV−1*) led to mild diarrhea, growth retardation, and damage to the intestinal mucosal villi [40]. They speculated that *PAstV* infection has a great influence on the intestinal mucosal injury. The results of this study revealed that *PAstV−4* infection activated ERK and MLC phosphorylation, followed by the down-regulation of MUC−2, occludin, and ZO−1 proteins. Moreover, the knockdown of NLRX1 hindered the activation of p−ERK and p−MLC and then alleviated intestinal mucosal barrier injury. We therefore hypothesized that endogenous NLRX1 might enhance *PAstV−4* replication and contribute to intestinal mucosal barrier function impairment via the ERK/MLCK pathway. ERK/MLCK is closely related to the regulation of intestinal barrier function [41,42]. In this study, we present the first report of NLRX1 playing a key role in regulating intestinal barrier injury through the ERK/MLCK pathway. However, whether this regulation is direct or indirect still needs to be further elucidated. The current research on NLRX1 mainly focuses on its effect on viral replication and mitophagy. It has been demonstrated that the interaction between NLRX1 and viral components might be a key factor in determining the outcome of viral infections [30,43]. In view of this, we guess that there is an intermedium between NLRX1 and the ERK/MLCK pathway, which will be further explored. The intermedium may be a certain viral protein of *PAstV−4* or a host protein in Caco−2 cells. Subsequently, screening and identification of the interacting proteins of NLRX1 will be carried out so as to resolve the specific mechanism of NLRX1 in the dysfunction of the intestinal mucosal barrier induced by *PAstV−4* infection.

There are five *PAstV* genotypes prevalent in the world with different clinical characteristics. Except *PAstV−3*, diarrhea is the main feature of all of the *PAstV* genotypes. *PAstV−4* is the most prevalent genotype worldwide. So, the data in this study can provide valuable clues for *PAstV* pathogenesis. However, the regulatory effects of NLRX1 on the other *PAstV* genotypes also need to be explored. 

In summary, this study first proposed that *PAstV−4* infection impaired the mucosal barrier function of Caco−2 cells by activating the ERK/MLCK pathway, and NLRX1 played a vital regulatory role in this network. These findings contribute to a better understanding of the clinical pathogenesis of *PAstV* infection and provide a foundation for future research on antiviral drug development. Importantly, exploring the mechanisms of intestinal mucosal barrier injury caused by different porcine diarrhea viruses contributes to finding the most common target molecules, so as to develop common antiviral drugs for the prevention and control of porcine diarrhea disease. 

## Figures and Tables

**Figure 1 cells-13-00913-f001:**
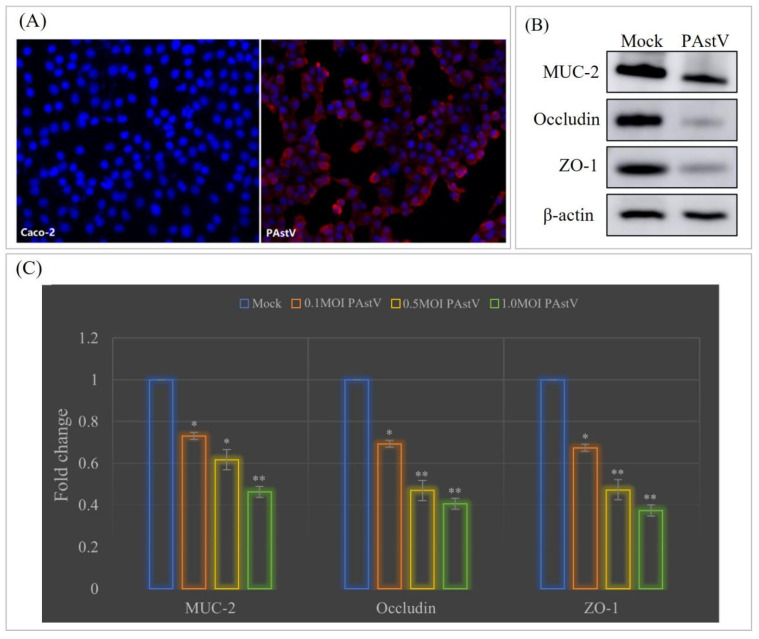
Down-regulation of tight junctions in Caco−2 cells upon *PAstV* infection. (**A**) *PAstV* propagation on Caco−2 cells with 15 μg/mL Pancreatin resulted in specific red fluorescence detected in the cytoplasm, as demonstrated by an indirect immunofluorescence assay using a *PAstV* monoclonal antibody. (**B**) Following inoculation of Caco-2 cells with 1MOI of *PAstV*, the expression of tight-junction proteins was detected 24 h later through Western blotting. (**C**) The influence of *PAstV* infection at various MOIs (0.1, 0.5, and 1.0) on the transcription of MUC−2, occludin, and ZO−1 was determined by relative fluorescence quantification PCR, utilizing *β-actin* as the intrinsic reference protein. Fold changes were calculated using the 2^−ΔΔCt^ method. (“**” *p* < 0.01, ”*” *p* < 0.05).

**Figure 2 cells-13-00913-f002:**
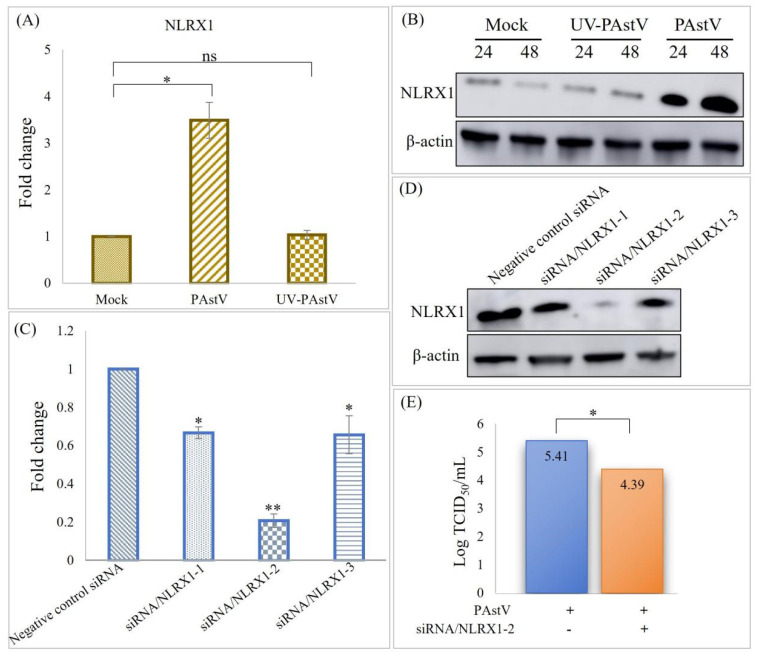
Reduction in *PAstV* replication through NLRX1 knockdown in Caco−2 cells. (**A**) Caco−2 cells were infected at an MOI of 1.0 with either *PAstV* or UV−inactivated *PAstV* for 24 h. Real−time PCR was employed to determine the relative NLRX1 levels, normalized to *β−actin*. Asterisks denote significant differences from uninfected cells(“**” *p* < 0.01, ”*” *p* < 0.05). “ns” indicates no significant difference. (**B**) Cell lysates were collected at 24 h and 48 h, respectively, following infection, and NLRX1 expression was assessed through Western blotting using the indicated antibodies. (**C**,**D**) Caco−2 cells were transfected with negative control siRNA or three different siRNA duplexes targeting NLRX1 for 24 h. Subsequently, the relative NLRX1 and expression levels were detected, with *β−actin* as an internal control. (**E**) Cells were transfected with negative control siRNA or siRNA/NLRX1−2 for 24 h and then infected with 1.0 MOI PAstV. Virus titers were determined through a TCID_50_ assay at 60 h post-infection.

**Figure 3 cells-13-00913-f003:**
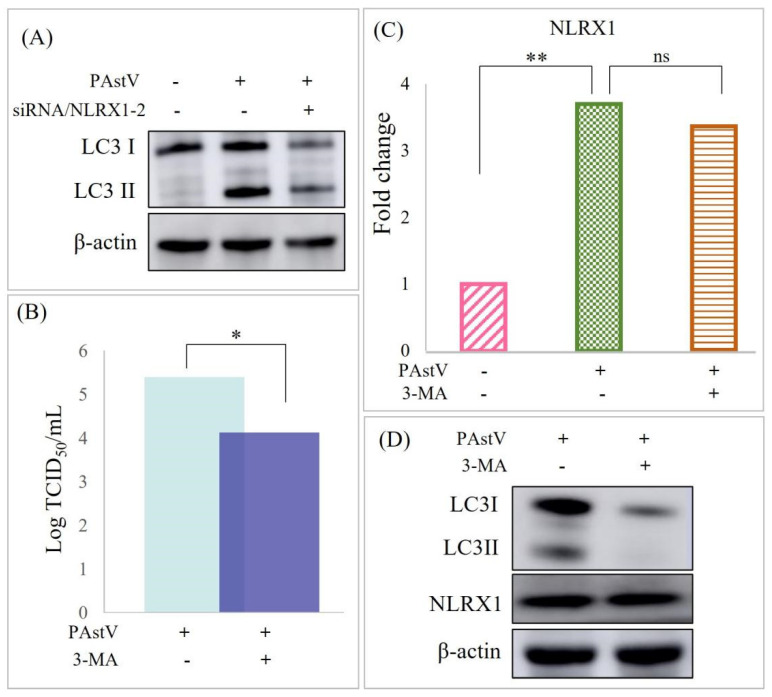
Induction of mitophagy by *PAstV* infection via up-regulation of NLRX1 protein. (**A**) Caco-2 cells were transfected with negative control siRNA or siRNA/NLRX1−2 for 24 h, followed by infection with 1MOI *PAstV*. Cell lysates were blotted with anti-LC3I/II, anti-NLRX1, and anti-*β−actin* antibodies. (**B**) Cells treated with the mitophagy inhibitor 3−MA for 6 h were subsequently infected with 1.0MOI *PAstV*. Virus titers were determined through a TCID_50_ assay 60 h post-infection. (**C**,**D**) show that cells subjected to treatment were infected with 1.0 MOI *PAstV*, and the relative and expression levels of NLRX1 were assessed 24 h later. (“**” *p* < 0.01, ”*” *p* < 0.05). “ns” indicates no significant difference.

**Figure 4 cells-13-00913-f004:**
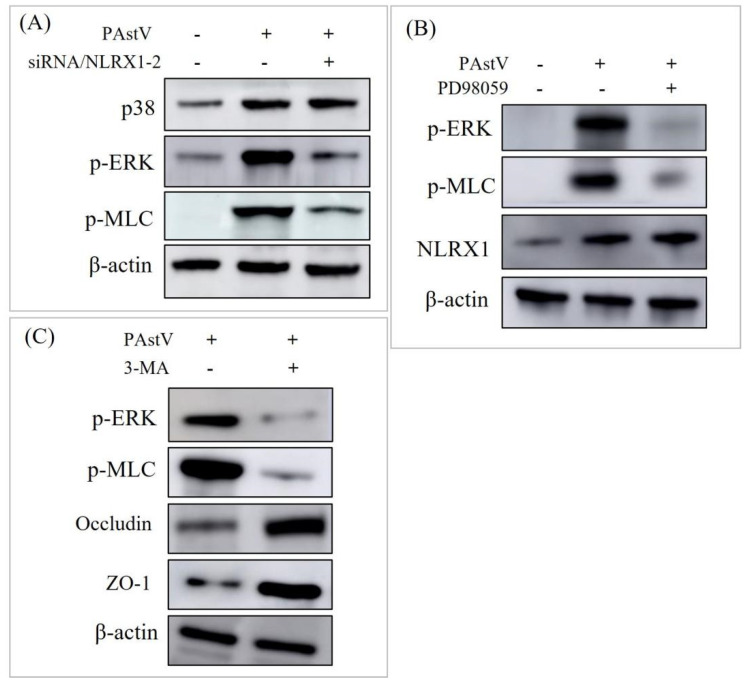
Reduction in tight-junction protein expression by *PAstV* infection via the ERK/MLC pathway. (**A**) Caco-2 cells were transfected with negative control siRNA or siRNA/NLRX1-2 for 24 h, followed by *PAstV* infection at an MOI of 1.0. After 24 h, cell lysates were blotted with the indicated antibodies. (**B**) Cells were treated with 20 μM PD98059 (ERK inhibitor) for 6 h, followed by infection with 1MOI *PAstV*. Subsequent Western blotting using the indicated antibodies was conducted as described above. (**C**) Cells were treated with 20 μM 3-MA (mitophagy inhibitor), and the subsequent steps were the same as in (**B**).

**Figure 5 cells-13-00913-f005:**
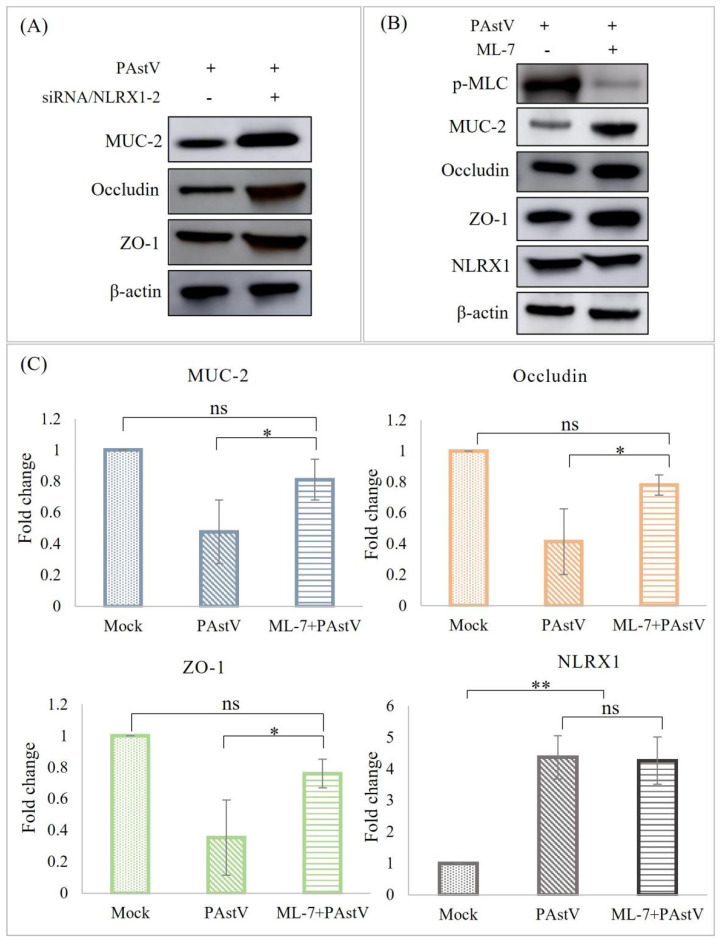
Essential role of NLRX1 in impaired intestinal barrier function induced by *PAstV* infection. (**A**) Caco−2 cells were transfected with negative control siRNA or siRNA/NLRX1−2 for 24 h, followed by *PAstV* infection at an MOI of 1.0. Cell lysates were then blotted with the indicated antibodies 24 h later. Further, cells were treated with 25 μM ML−7 (MLCK inhibitor) for 6 h and infected with 1.0 MOI *PAstV*. (**B**) The relative MUC−2, occludin, ZO−1, and NLRX1 levels were detected after treatment with ML−7 (MLCK inhibitor), using *β−actin* as an internal reference. (**C**) MUC−2, occludin, ZO−1, and NLRX1 expression levels were assessed after treatment with ML−7 (MLCK inhibitor), with *β−actin* as an internal control. (“**” *p* < 0.01, ”*” *p* < 0.05). “ns” indicates no significant difference.

## Data Availability

The data are contained within the article.

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
