# Peer review of "NLRX1 Mediates the Disruption of Intestinal Mucosal Function Caused by Porcine Astrovirus Infection via the Extracellular Regulated Protein Kinases/Myosin Light–Chain Kinase (ERK/MLCK) Pathway"

_cells, 2024, doi:10.3390/cells13110913_

Round 1
Reviewer 1 Report
Comments and Suggestions for Authors
The paper by Tao et al studies the role of NLRX1 in disruption of intestinal mucosal function after porcine astrovirus infection.
I would advise the authors to revise the manuscript in several sections.
In the introduction, the level of detail on porcine astroviruses biology is very limited. There are five porcine astroviruses, pAst1-5, and these astroviruses do not necessarily share a common biology. In literature, pAst3 infections are described relatively early in life, around 3 weeks of age, even in clinically healthy piglets. Other astroviruses infect directly before or after weaning (pAstV4 also). The virus is found in both clinically healthy and in diseased piglets. It is of note that PEDV infects usually much earlier in the life of the piglets. The authors make reference to astrovirus diversity in line 38, but the description is vague and I am not sure if all astrovirus genotypes have been tested in experimental model to study their exact pathology.
Also the statement on recombination in line 40 is vague and without references. Have pathogenic astroviruses that arose from recombination events between human and porcine astrovirus been described?
I would encourage the authors to check if all astroviruses infect intestinal epithelial cells only. There are other cell types in the gut that may be infected, and to my knowledge infection of Goblet cells has been described for Astrovirus 3 in mice.
A suggestion would be to mention throughout the manuscript that the study is specifically conducted on pAstV4.
The discussion is relatively short, but context should be given if the finding on NLRX1 will be relevant to all porcine astroviruses
In the discussion, PAstV-GX1 is mentioned, but it is unclear to the reader what genotype this is.
The description of material and methods and results appears adequate.
The primary data in the paper show consistent results.
In the discussion there is more room to go in depth on context on pAstrV4 available in literature. I miss that. The discussion is very concise and I have the feeling that references to relevant primary papers that study pAstV4 is missing.
Comments on the Quality of English LanguageNo comments on English language (I am not a native speaker myself)
Reviewer 2 Report
Comments and Suggestions for Authors
In this study, Tao et al. have demonstrated that PAstV-4 infection can lead to impairment of the intestinal mucosal barrier, including the down-regulation of tight junction proteins MUC-2, Occludin, and ZO-1. Porcine astrovirus (PAstV) is a potential zoonotic pathogen that can cause symptoms such as diarrhea and neurological issues in various species, including pigs. The NLRX1 protein, predominantly found in mitochondria, is upregulated following PAstV-4 infection and may play a crucial role in PAstV infection. This study concludes that activation of the ERK/MLCK pathway by PAstV-4 infection can contribute to the impairment of the mucosal barrier function in Caco-2 cells. However, the logical connection between NLRX1 and mucosal barrier function is lacking and needs further elucidation.
Some minor comments:
· A bit more explanation of the NLRX1-involved ERK/MLCK signaling pathway would be necessary for readers to better understand the mechanism. It would be helpful to provide a graph abstract to demonstrate the hypothetical model to clarify the mechanism of NLRX1 upregulation following PAstV-4 infection.
· Explore the clinical implications of the findings, such as the development of antiviral drugs or strategies to mitigate the impairment of the mucosal barrier function caused by PAstV infection.
· Discuss the relevance of the findings to other astroviruses and compare the impact of different astrovirus strains on the mucosal barrier function.
· Consider future research directions based on the findings.
Comments on the Quality of English LanguageNA
